# Revisiting the Role of Homophily in Fair Graph Representation Learning

## Abstract

Graph Neural Networks (GNNs) can propagate sensitive signals via message passing, especially on homophilous graphs where edges preferentially connect nodes sharing sensitive attributes. We revisit fairness through the lens of homophily using CSBM-S, a synthetic model that independently controls label homophily $h_y$ and sensitive homophily $h_s$, enabling precise and repeatable evaluation of group fairness. CSBM-S reveals two key observations: (i) group disparity peaks when $h_y \approx 0.5$; and (ii) bias consistently diminishes as $h_s \to 0.5$. Guided by these insights, we propose FairEST, which enforces $h_s \approx 0.5$ by flipping the sensitive attribute and its most correlated features during training. Across diverse benchmarks and backbones, FairEST attains the lowest bias on most encoder-dataset pairs with comparable accuracy, yielding average absolute reductions of $1.63\%$ ($\Delta$SP) and $1.28\%$ ($\Delta$EO) over the prior state-of-the-art. Together, CSBM-S and FairEST provide a homophily-centric toolkit for analyzing and mitigating bias in graph representation learning.

## 1 Introduction

Fairness is a central concern in machine learning: models can produce systematically different predictions across groups defined by sensitive attributes (Lambrecht & Tucker, 2019; Mehrabi et al., 2021; Barocas et al., 2023). For example, consider a model trained to select national-team athletes. If historical rosters are mostly composed of players under 30, the model may learn this spurious correlation between "age $< 30$" and "selection," thereby disfavoring candidates aged $\geq 30$ even if their skills are equal or better. Here, age is the sensitive attribute partitioning individuals into two groups ($< 30$ vs. $\geq 30$), and the predictions are said to be biased with respect to age. Beyond this example, such disparities can exacerbate inter-group conflicts and create social harms (Selbst et al., 2019; Bender et al., 2021; Angwin et al., 2022).

In fact, Graph Neural Networks (GNNs) are particularly prone to fairness issues. Message passing couples node features with graph topology, typically improving utility but also pushing adjacent nodes toward similar embeddings, a phenomenon called over-smoothing (Oono & Suzuki, 2019). When a graph exhibits **sensitive homophily**—edges more likely connect nodes sharing a sensitive attribute—sensitive signals are propagated throughout the graph via message passing, turning local neighborhoods into proxies for group membership. Consequently, even mild feature-level biases can be amplified into severe representation-level disparities (Wang et al., 2022).

Existing debiasing approaches largely fall into two categories: (i) general-purpose techniques that enforce invariance to the sensitive attribute—e.g., adversarial training (Dai & Wang, 2021); and (ii) propagation-aware interventions that modify message passing, such as masking sensitive channels (Wang et al., 2022) or injecting neutralizing features (Yang et al., 2024). While often effective, these methods typically rely on auxiliary modules and introduce numerous hyperparameters, making them sensitive to tuning and harder to apply.

**This paper revisits homophily as the driver of bias amplification in GNNs.** We ask: *How do label homophily and sensitive homophily—denoted $h_y$ and $h_s$—shape fairness under message passing?* To answer this question, we introduce **CSBM-S**, a synthetic graph model that independently controls $h_y$ and $h_s$. Using CSBM-S, we find two observations: (1) bias peaks near $h_y \approx 0.5$, and is further amplified under extreme sensitive homophily (e.g., $h_s \approx 0.1$ or $0.9$); and (2) bias decreases as $h_s \to 0.5$, with this dependence weakening as class-related features become more informative.

Building on these insights, we propose **FairEST**, a fairness-aware GNN that **Enforces SensiTive homophily to 0.5**. FairEST iteratively adjusts sensitive attributes and their most correlated features to steer each node's neighborhood toward a balanced (50-50) mix of same- vs. different-attribute neighbors, thereby limiting the propagation of sensitive signals during message passing. On diverse encoder-dataset pairs, FairEST substantially improves group fairness while maintaining accuracy comparable to vanilla GNNs.

**Contributions.** To summarize, our contributions are as follows:

- We introduce **CSBM-S**, a synthetic graph model with tunable label/sensitive homophily, enabling precise and repeatable evaluation of fairness under message passing.
- Using CSBM-S, we analyze when message passing amplifies vs. attenuates disparities, identifying $h_s$ as the primary driver and clarifying its interaction with $h_y$.
- We propose **FairEST**, which enforces $h_s \approx 0.5$ by iteratively flipping sensitive attributes and their most correlated features.
- On multiple encoder-dataset pairs, we show that FairEST substantially improves fairness with accuracy on par with vanilla GNNs.

## 2 RELATED WORK

Fairness-aware GNNs aim to mitigate group disparities while preserving utility. FairGNN (Dai & Wang, 2021) applies adversarial learning to learn representations invariant to the sensitive attribute. NIFTY (Agarwal et al., 2021) augments the training objective with a fairness regularizer term, and employs layer-wise weight normalization. EDITS (Dong et al., 2022) decomposes debiasing into attribute- and structure-level components, updating both the feature matrix and the adjacency. FairVGNN (Wang et al., 2022) masks sensitive-related channels and applies adaptive weight clamping to restrict propagation. FairSIN (Yang et al., 2024) injects Fairness-Facilitating Features (F3) that neutralize sensitive signals while emphasizing non-sensitive information.

## 3 PRELIMINARIES

### 3.1 NOTATION

Let $\mathcal{G} = (\mathcal{V}, \mathcal{E})$ be an undirected graph with $n = |\mathcal{V}|$ nodes. The node feature matrix is $X \in \mathbb{R}^{n \times d}$. The adjacency matrix $A \in \{0, 1\}^{n \times n}$ satisfies $A_{u,v} = 1$ iff $(u, v) \in \mathcal{E}$. For node $v$, denote its neighborhood by $\mathcal{N}(v) = \{u \in \mathcal{V} : A_{u,v} = 1\}$ and degree by $d_v = |\mathcal{N}(v)|$.

We consider binary node classification with labels $\mathbf{y} \in \{0, 1\}^n$ and a binary sensitive attribute $\mathbf{s} \in \{0, 1\}^n$. Unless otherwise noted, $s$ is observed for all nodes (train/val/test). A fairness-aware GNN learns a predictor $f_\theta : (X, A) \to \hat{\mathbf{y}}$ that outputs label predictions, aiming for high accuracy while minimizing the statistical dependence of $\hat{y}$ on $s$.

### 3.2 HOMOPHILY MEASURES

Homophily quantifies the tendency of adjacent nodes to share an attribute (Abu-El-Haija et al., 2019; Attali et al., 2024). For a node $v$, the node-level label homophily (Pei et al., 2020) is

$$h_y(v) = \frac{1}{d_v} \sum_{u \in \mathcal{N}(v)} \mathbf{1}_{\mathbf{y}_u = \mathbf{y}_v}. \tag{1}$$

Analogously, for a binary sensitive attribute $s$, the node-level sensitive homophily is

$$h_s(v) = \frac{1}{d_v} \sum_{u \in \mathcal{N}(v)} \mathbf{1}_{\mathbf{s}_u = \mathbf{s}_v}. \tag{2}$$

Intuitively, high $h_y$ indicates label-consistent neighborhoods, so message passing aggregates task-relevant signals and typically improves accuracy (Huang et al., 2023; Peng et al., 2024). By contrast, high $h_s$ indicates neighborhoods segregated by the sensitive attribute, so message passing propagates sensitive information and thereby degrades fairness.

### 3.3 FAIRNESS METRICS

Fairness is commonly distinguished as group vs. individual fairness. Group fairness requires comparable predictions across sensitive groups, whereas individual fairness requires similar individuals to receive similar predictions (Kang et al., 2023). We focus on group fairness, which is standard in the GNN literature (Hussain et al., 2022; Zhang et al., 2023; Luo et al., 2024) and aligned with our evaluation protocol.

Let $s$ denote the sensitive attribute and $\hat{y}$ the model's binary predictions. We quantify the dependence of $\hat{y}$ on $s$ using two group-fairness metrics:

- **Statistical parity** (Dwork et al., 2012) measures disparity in positive prediction rates:
$$\Delta\text{SP} = |P(\hat{y} = 1|s = 0) - P(\hat{y} = 1|s = 1)|. \tag{3}$$

- **Equal opportunity** (Hardt et al., 2016) measures disparity in true positive rates:
$$\Delta\text{EO} = |P(\hat{y} = 1|y = 1, s = 0) - P(\hat{y} = 1|y = 1, s = 1)|. \tag{4}$$

Both metrics lie in $[0, 1]$ (lower is better), with $0$ indicating no measured group disparity. Throughout this paper, we report absolute gaps as percentage points ($\times 100\%$).

## 4 CSBM-S

### 4.1 OVERVIEW

Contextual stochastic block models (CSBMs) are used for generating synthetic graphs—including node features and adjacency—providing controlled testbeds for graph representation learning (Deshpande et al., 2018; Luan et al., 2023; Lee et al., 2024). We extend CSBM to **CSBM-S**, which injects a tunable sensitive attribute into the feature-generation process. CSBM-S offers explicit control over label and sensitive homophily, enabling precise and repeatable evaluation of group fairness under message passing.

### 4.2 CSBM-S DEFINITION

A CSBM-S instance is specified by the following components.

- **Nodes** ($n$). Let $n$ be even, with balanced classes: $P(y = 0) = P(y = 1) = \frac{1}{2}$.
- **Label-sensitive correlation** ($\rho$). Assume balanced sensitive groups, $P(s = 0) = P(s = 1) = \frac{1}{2}$. Let $\rho \in [-1, 1]$ control the correlation between $\mathbf{y}$ and $\mathbf{s}$, with joint distributions:
$$P(y = 0, s = 0) = P(y = 1, s = 1) = \frac{1 + \rho}{4},$$
$$P(y = 0, s = 1) = P(y = 1, s = 0) = \frac{1 - \rho}{4}. \tag{5}$$

- **Feature generation.** Each node has two scalar features $(x_y, x_s) \in \mathbb{R}^2$. Both features are sampled from Gaussians, conditioned on $y$ and $s$ respectively:
$$x_y \sim \begin{cases} \mathcal{N}(\mu_{y0}, \sigma_{y0}^2), & y = 0, \\ \mathcal{N}(\mu_{y1}, \sigma_{y1}^2), & y = 1, \end{cases} \qquad x_s \sim \begin{cases} \mathcal{N}(\mu_{s0}, \sigma_{s0}^2), & s = 0, \\ \mathcal{N}(\mu_{s1}, \sigma_{s1}^2), & s = 1. \end{cases} \tag{6}$$

Then, label and sensitive feature gaps are defined as
$$\Delta_y = |\mu_{y1} - \mu_{y0}|, \quad \Delta_s = |\mu_{s1} - \mu_{s0}|. \tag{7}$$

- **Neighborhood composition.** Assume a global neighborhood composition shared by all nodes, with degree $d = d_+^+ + d_-^+ + d_+^- + d_-^-$. The superscript indicates whether a neighbor shares the label $y$ ($+$ same, $-$ different), and the subscript indicates whether it shares the sensitive attribute $s$. Then, label and sensitive homophily can be computed as
$$h_y = \frac{d_+^+ + d_-^+}{d}, \quad h_s = \frac{d_+^+ + d_+^-}{d}. \tag{8}$$

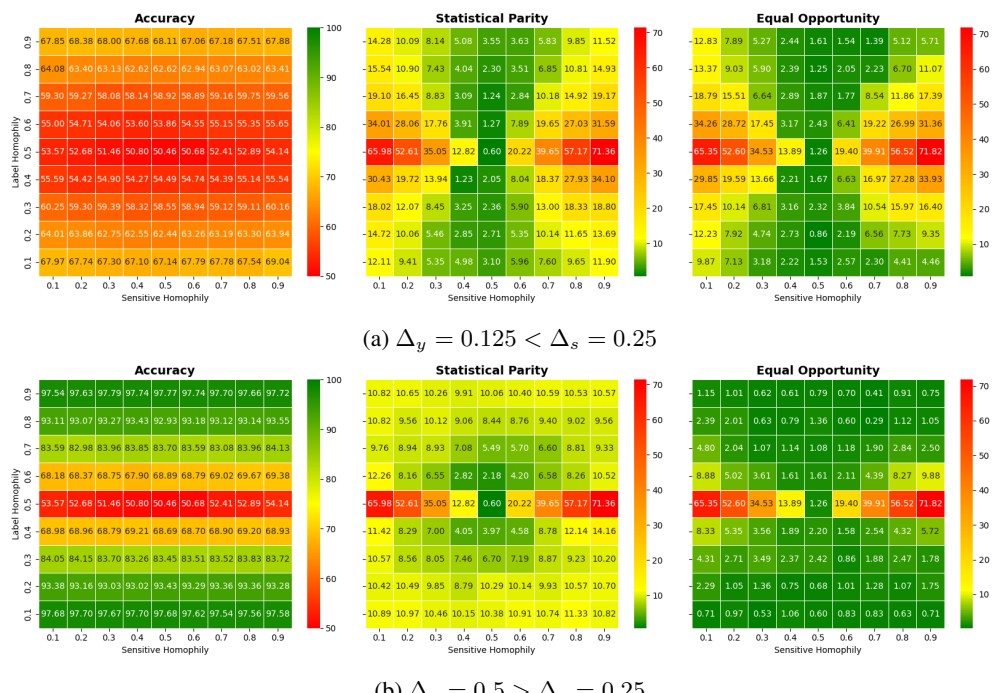

(a) $\Delta_y = 0.125 < \Delta_s = 0.25$

(b) $\Delta_y = 0.5 > \Delta_s = 0.25$

Figure 1: Heatmaps of accuracy (left), $\Delta$SP (middle), and $\Delta$EO (right) over a grid of label and sensitive homophily $(h_y, h_s) \in \{0.1, 0.2, \dots, 0.9\}^2$. The sensitive feature gap is fixed at $\Delta_s = 0.25$. Each subfigure corresponds to label feature gaps of: (a) $\Delta_y = 0.125$ and (b) $\Delta_y = 0.5$. Fairness metrics peak near $h_y \approx 0.5$ and diminish as $h_s \to 0.5$. As $\Delta_y$ increases, dependence on $h_s$ weakens, shifting the dominant orientation from vertical in (a) to horizontal in (b).

### 4.3 CSBM-S EXPERIMENT

We examine how homophily shapes accuracy and group fairness.

**Homophily sweep.** We sweep $(h_y, h_s) \in \{0.1, 0.2, \dots, 0.9\}^2$. For each grid cell, we instantiate a CSBM-S graph following Section 4.2 with the same $n$, $\rho$, and $d$.

**Metrics.** Under a fixed training setting, we report node-classification accuracy and the two group-fairness metrics from Section 3.3: statistical parity ($\Delta$SP) and equal opportunity ($\Delta$EO).

**Settings.** We fix the sensitive feature gap $\Delta_s = 0.25$ and consider two label feature gaps: weak task signals $\Delta_y = 0.125$ and strong task signals $\Delta_y = 0.5$. Heatmaps (Fig. 1a-b) summarize accuracy, $\Delta$SP, and $\Delta$EO across the grid.

**Empirical trends.**

- **Accuracy.** Accuracy is minimal when $h_y \approx 0.5$ (strong class mixing) and increases as $|h_y - 0.5|$ grows. Sensitivity to $h_s$ is minor. Larger $\Delta_y$ further improves accuracy.

- **$\Delta$SP and $\Delta$EO.** Both peak at $h_y \approx 0.5$ and diminish as $h_s \to 0.5$. As $\Delta_y$ increases, dependence on $h_s$ weakens, shifting the dominant orientation from vertical to horizontal (i.e., from $h_s$-driven to $h_y$-driven).

### 4.4 MAIN OBSERVATIONS

From the results in Section 4.3, we highlight two observations: (i) bias is largest when $h_y \approx 0.5$, and is further amplified under extreme sensitive homophily (e.g., $h_s \approx 0.1$ or $0.9$); (ii) bias decreases as $h_s \to 0.5$, with this dependence weakening when class-related features dominate (e.g., $\Delta_y > \Delta_s$). We now provide a theoretical analysis that explains these trends.

**Initial features.** For simplicity, assume symmetric means and unit variances: $\mu_y = \mu_{y1} = -\mu_{y0}$, $\mu_s = \mu_{s1} = -\mu_{s0}$, and $\sigma_{y0} = \sigma_{y1} = \sigma_{s0} = \sigma_{s1} = 1$. Let $\varphi(x; \mu, \sigma^2) = \frac{1}{\sqrt{2\pi}\sigma} \exp\left(-\frac{(x-\mu)^2}{2\sigma^2}\right)$. Then, the joint pdf of initial features $(x_y, x_s)$ is the four-component mixture:

$$f_{x_y,x_s}(x_y, x_s) = \frac{1+\rho}{4}\varphi(x_y; \mu_y, 1)\varphi(x_s; \mu_s, 1) + \frac{1-\rho}{4}\varphi(x_y; \mu_y, 1)\varphi(x_s; -\mu_s, 1)$$
$$+ \frac{1-\rho}{4}\varphi(x_y; -\mu_y, 1)\varphi(x_s; \mu_s, 1) + \frac{1+\rho}{4}\varphi(x_y; -\mu_y, 1)\varphi(x_s; -\mu_s, 1). \tag{9}$$

**Aggregated features.** Following the mean-field one-hop aggregation model of Lee et al. (2024), the aggregated features $(z_y, z_s)$ satisfy

$$z_y \mid y = 1 \sim \mathcal{N}\left(\mu_y(2h_y - 1), 1/d\right), \quad z_y \mid y = 0 \sim \mathcal{N}\left(-\mu_y(2h_y - 1), 1/d\right),$$
$$z_s \mid s = 1 \sim \mathcal{N}\left(\mu_s(2h_s - 1), 1/d\right), \quad z_s \mid s = 0 \sim \mathcal{N}\left(-\mu_s(2h_s - 1), 1/d\right), \tag{10}$$

where $h_y$ and $h_s$ denote label and sensitive homophily, and $d$ is the degree. The joint pdf remains a four-component mixture with homophily-shifted means:

$$f_{z_y,z_s}(z_y, z_s) = \frac{1+\rho}{4} \varphi(z_y; \mu_y(2h_y - 1), 1/d) \varphi(z_s; \mu_s(2h_s - 1), 1/d)$$
$$+ \frac{1-\rho}{4} \varphi(z_y; \mu_y(2h_y - 1), 1/d) \varphi(z_s; -\mu_s(2h_s - 1), 1/d)$$
$$+ \frac{1-\rho}{4} \varphi(z_y; -\mu_y(2h_y - 1), 1/d) \varphi(z_s; \mu_s(2h_s - 1), 1/d)$$
$$+ \frac{1+\rho}{4} \varphi(z_y; -\mu_y(2h_y - 1), 1/d) \varphi(z_s; -\mu_s(2h_s - 1), 1/d). \tag{11}$$

**Observation 1.** When $h_y = 0.5$, the class-aligned mean $\pm\mu_y(2h_y - 1)$ is zero, so $z_y$ carries no discriminative signal and predictions must rely on the sensitive channel $z_s$. As separation along $z_s$ increases—i.e., as $|\mu_s(2h_s - 1)|$ grows—$\Delta$SP and $\Delta$EO increase, inflating group disparity.

**Observation 2.** Sensitive-channel separability scales with $|\mu_s(2h_s - 1)|$. Driving $h_s \to 0.5$ reduces this gap, making predictions less dependent on $s$. This effect weakens when class signals dominate ($\Delta_y > \Delta_s$), since the classifier places more weight on $z_y$ than on $z_s$.

**Interpretation.** After message passing, label homophily and sensitive homophily control the effective signal strengths of the task and sensitive channels, respectively. Since bias is maximized when the propagated task signal is weak relative to the propagated sensitive signal, Figure 1 exhibits larger disparities when $h_y \approx 0.5$ and $h_s \approx 0.1$ or $0.9$.

## 5 PROPOSED METHOD: FAIREST

### 5.1 OVERVIEW

Guided by the observations in Section 4.4, we propose **FairEST**, a fairness-aware GNN that mitigates prediction bias by **Enforcing SensiTive homophily to 0.5**. We prioritize steering $h_s \to 0.5$ (Obs. 2) over altering $h_y$ (Obs. 1) because: (i) modifying label homophily can harm accuracy (Fig. 1); and (ii) in node classification, ground-truth test labels are unavailable whereas sensitive attributes are typically observable during training (Yang et al., 2024). FairEST enforces $h_s \approx 0.5$ by selectively flipping sensitive attributes and their most correlated features during training. Section 5.2 details the algorithmic pipeline.

### 5.2 FAIREST ALGORITHM

For each node $v$ in a random order $\pi$, compute

$$\text{Same}(v) = |\{u \in \mathcal{N}(v) : \mathbf{s}_u = \mathbf{s}_v\}|,$$
$$\text{Diff}(v) = |\{u \in \mathcal{N}(v) : \mathbf{s}_u \neq \mathbf{s}_v\}|, \tag{12}$$

and the number of updates to apply in $v$'s neighborhood,

$$\text{Flip}(v) = \lfloor |\text{Same}(v) - \text{Diff}(v)| / 2 \rfloor. \tag{13}$$

If $\text{Flip}(v) > 0$, form the majority set

$$\mathbb{M}(v) = \begin{cases} \{u \in \mathcal{N}(v) : \mathbf{s}_u = \mathbf{s}_v\}, & \text{Same}(v) > \text{Diff}(v), \\ \{u \in \mathcal{N}(v) : \mathbf{s}_u \neq \mathbf{s}_v\}, & \text{otherwise}. \end{cases} \tag{14}$$

Sample Flip$(v)$ nodes $u \in \mathbb{M}(v)$ without replacement, and update each selected neighbor as follows.

**Sensitive attribute flip.**

$$\mathbf{s}_u \leftarrow 1 - \mathbf{s}_u. \tag{15}$$

**Correlated feature reflection.** Let $\mathbb{J}_k \subseteq \{1, \ldots, d\}$ denote the set of $k$ feature dimensions with the largest absolute Pearson correlation $|\mathrm{Corr}(X_{:,j}, \mathbf{s})|$. For each $j \in \mathbb{J}_k$, apply

$$X_{u,j} \leftarrow (\mathbf{x}_j^{\max} + \mathbf{x}_j^{\min}) - X_{u,j}, \tag{16}$$

where $\mathbf{x}_j^{\max}$ and $\mathbf{x}_j^{\min}$ are the per-feature max/min. This mid-range reflection inverts the sensitive signal on those coordinates while preserving scale.

We repeat this while the average sensitive homophily $\bar{h}_s = \frac{1}{|\mathcal{V}|} \sum_{v \in \mathcal{V}} h_s$ moves closer to 0.5 (i.e., $|\bar{h}_s - 0.5|$ decreases), up to $T_{\max} = 10$ iterations. The edited $(\mathbf{s}, X)$ are used only during training; inference uses the original graph and features.

## 5.3 FAIRNESS LOSS

To complement homophily enforcement, we add a group-fairness regularizer that penalizes residual dependence between predictions and the sensitive attribute (Luo et al., 2024). Let $\hat{p}_\theta(v)$ denote the predicted positive-class probability for node $v$. We target statistical parity (SP) and equal opportunity (EO) via group-conditional mean differences:

$$
\begin{aligned}
\mathrm{SP}(\theta) &= |\, \mathbb{E}[\hat{p}_\theta(v) \mid s = 0] - \mathbb{E}[\hat{p}_\theta(v) \mid s = 1] \,|, \\
\mathrm{EO}(\theta) &= |\, \mathbb{E}[\hat{p}_\theta(v) \mid y = 1,\ s = 0] - \mathbb{E}[\hat{p}_\theta(v) \mid y = 1,\ s = 1] \,|.
\end{aligned} \tag{17}
$$

The training objective augments the supervised loss with a fairness penalty:

$$\mathcal{L}(\theta) = \mathcal{L}_{cls}(\theta) + \lambda(\mathrm{SP}(\theta) + \mathrm{EO}(\theta)), \tag{18}$$

where $\mathcal{L}_{cls}$ is the cross-entropy loss of labeled nodes, and $\lambda \geq 0$ controls the regularization strength. Expectations are computed over the current mini-batch (or the training split). This regularizer term suppresses remaining disparities in predictions, reinforcing the push toward $h_s \approx 0.5$.

## 6 EXPERIMENT

We address the following research questions:

- **RQ1 (Effectiveness).** How well does FairEST reduce group-level bias? (Section 6.2)
- **RQ2 (Sensitivity to $k$).** How does the number of correlated features modified by FairEST affect performance? (Section 6.3)
- **RQ3 (Ablations).** What are the individual contributions of the pre-processing algorithm and the fairness loss? (Section 6.4)
- **RQ4 (Noisy sensitive attributes).** How robust is FairEST when sensitive attributes are noisy or partially observed? (Section 6.5)

## 6.1 EXPERIMENTAL SETUP

**Datasets.** We evaluate on three real-world benchmarks—German, Credit, and Bail (Wang et al., 2022). **German**: nodes are bank clients; edges connect clients with similar account profiles; task is high/low credit-risk classification; sensitive attribute is gender. **Credit**: nodes are credit-card users; edges connect users with similar purchase/payment patterns; task is default prediction; sensitive attribute is age. **Bail**: nodes are defendants released between 1990 and 2009; edges link defendants with similar criminal histories and demographics; task is recidivism prediction after release; sensitive attribute is race. Additional statistics appear in Appendix B.2.

**Backbones & baselines.** We evaluate FairEST on three GNN backbones—GCN (Kipf & Welling, 2016), GIN (Xu et al., 2018), and GraphSAGE (Hamilton et al., 2017)—and compare against fairness-aware baselines: FairGNN (Dai & Wang, 2021), NIFTY (Agarwal et al., 2021), EDITS (Dong et al., 2022), FairVGNN (Wang et al., 2022), and FairSIN (Wang et al., 2022). Implementation details, including hyperparameters and data splits, are provided in Appendix C.3.

Table 1: Node classification on German, Credit, and Bail, using GCN, GIN, and GraphSAGE backbones. We report accuracy (ACC↑) and group-fairness—statistical parity (SP↓) and equal opportunity (EO↓). Within each encoder–dataset pair, **bold** marks the highest ACC and the lowest SP/EO. Values are means over 5 seeds; standard deviations are reported in Appendix D.1.

| Encoder | Method | German | | | Credit | | | Bail | | |
|---|---|---|---|---|---|---|---|---|---|---|
| | | ACC↑ | SP↓ | EO↓ | ACC↑ | SP↓ | EO↓ | ACC↑ | SP↓ | EO↓ |
| GCN | Vanilla | 70.00 | 0.17 | 0.19 | 74.94 | 8.18 | 6.50 | 86.15 | 5.50 | 3.32 |
| | FairGNN | 69.92 | 0.29 | 0.57 | 74.67 | 4.95 | 2.43 | 86.62 | 7.18 | 4.59 |
| | NIFTY | 70.24 | 1.31 | 0.99 | 73.19 | 9.01 | 7.11 | 80.32 | 4.30 | 3.49 |
| | EDITS | **70.32** | 1.35 | 0.61 | 71.83 | 7.95 | 5.60 | 75.57 | 7.69 | 6.15 |
| | FairVGNN | 70.16 | 1.44 | 1.30 | **78.38** | 3.59 | 1.68 | 85.50 | 6.65 | 5.07 |
| | FairSIN | 70.16 | 0.51 | 0.50 | 77.86 | 0.50 | 0.33 | **87.68** | 5.45 | 4.18 |
| | FairEST | 70.16 | **0.08** | **0.15** | 77.86 | **0.03** | **0.03** | 85.36 | **0.85** | **0.36** |
| GIN | Vanilla | 71.04 | 3.36 | 1.26 | 75.73 | 1.18 | 0.66 | 84.81 | 7.27 | 6.55 |
| | FairGNN | **71.28** | 2.38 | 0.51 | 74.31 | 2.26 | 1.00 | 86.43 | 6.18 | 3.65 |
| | NIFTY | 70.00 | **0.08** | **0.17** | 75.40 | 3.31 | 2.92 | 86.05 | 5.57 | 4.34 |
| | EDITS | 71.04 | 1.99 | 1.12 | 74.36 | 5.85 | 4.08 | 72.33 | 4.15 | 4.33 |
| | FairVGNN | 71.20 | 3.27 | 1.30 | 77.31 | 1.08 | 0.97 | **88.19** | 6.32 | 5.40 |
| | FairSIN | 70.72 | 2.06 | 1.72 | 77.88 | 0.06 | **0.04** | 86.25 | 4.67 | 4.42 |
| | FairEST | 70.08 | 0.21 | **0.17** | **78.26** | 0.05 | 0.31 | 82.71 | **1.70** | **0.94** |
| SAGE | Vanilla | 70.00 | 0.85 | 0.71 | 74.30 | 8.68 | 6.44 | 88.21 | 0.67 | 1.92 |
| | FairGNN | 69.28 | 1.07 | 1.77 | 76.11 | 2.56 | 1.64 | 85.63 | **0.65** | **1.59** |
| | NIFTY | 69.68 | 0.67 | 1.62 | 75.98 | 8.09 | 5.82 | 87.02 | 4.05 | 2.30 |
| | EDITS | 71.04 | 4.80 | 2.08 | 73.08 | 11.24 | 8.99 | 75.18 | 5.99 | 6.47 |
| | FairVGNN | 71.20 | 5.56 | 2.14 | **79.54** | 6.30 | 3.78 | 88.41 | 3.76 | 1.76 |
| | FairSIN | 70.88 | 4.04 | 2.18 | 78.65 | 1.96 | 0.85 | **88.58** | 1.14 | 2.19 |
| | FairEST | **71.28** | **0.25** | **0.15** | 77.86 | **0.06** | **0.12** | 87.44 | 2.46 | 2.63 |

## 6.2 MAIN RESULTS

Table 1 reports utility and fairness across encoders and datasets. FairEST achieves the lowest bias on most encoder-dataset pairs, yielding average absolute reduction of $1.63\%$ ($\Delta$SP) and $1.28\%$ ($\Delta$EO) over the previous state-of-the-art (FairSIN). We attribute these gains to FairEST's enforcement of balanced exposure to sensitive information during message passing, which limits the propagation of sensitive signals. Improvements are most pronounced when the vanilla backbone exhibits substantial disparities, such as SAGE-Credit and GIN-Bail. In contrast, when baseline bias is relatively small (e.g., SAGE-Bail), gains are modest and may be affected by shifts in the marginal distribution $P(s)$. In terms of utility, FairEST remains competitive with the vanilla models, typically incurring only minor accuracy changes ($-2.1\%$ in GIN-Bail $\sim +3.6\%$ in SAGE-Credit). Overall, FairEST delivers substantial group-fairness improvements while maintaining comparable predictive performance.

## 6.3 HYPERPARAMETER STUDY

We analyze sensitivity to $k$, the number of feature dimensions that FairEST modifies. Increasing $k$ expands the perturbation budget, whereas $k = 0$ updates only the sensitive attribute. While sweeping $k \in \{0, 1, 2, 3, 4\}$, we plot accuracy and a composite bias score ($\Delta$SP + $\Delta$EO).

Figure 2 (GIN encoder) reveals two trends. First, the $k$ value that minimizes bias is dataset-specific: bias roughly decreases with $k$ on Credit, but rises sharply on German and Bail when $k = 0 \to 1$. We attribute this to variation across datasets in the number of feature dimensions that can be considered "correlated." Second, accuracy is stable across $k$, except on Bail, which drops by over $20\%$ when $k = 3 \to 4$. This suggests over-perturbation, whereby large $k$ perturbs not only sensitive-related but also class-related features, restricting the propagation of task signals. In practice, $k \le 3$ offers a reasonable operating regime that balances bias reduction with stable accuracy.

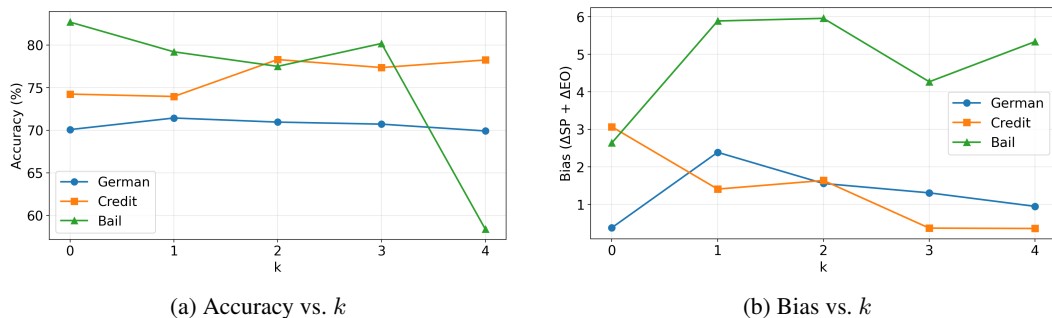

(a) Accuracy vs. $k$             (b) Bias vs. $k$

Figure 2: Sensitivity to $k$—the number of feature dimensions modified by FairEST—on the GIN encoder. We sweep $k \in \{0, 1, 2, 3, 4\}$ and report (a) accuracy and (b) composite bias $(\Delta \text{SP} + \Delta \text{EO})$. Bias curves are dataset-specific (e.g., decreases with $k$ on Credit; rises sharply on German/Bail when $k = 0 \rightarrow 1$). Accuracy is stable, with a notable drop on Bail when $k = 3 \rightarrow 4$.

Table 2: Ablation of FairEST components on German, Credit, and Bail using GCN, GIN, and Graph-SAGE backbones. Reported metrics are accuracy (ACC↑) and fairness—statistical parity (SP↓) and equal opportunity (EO↓). Within each encoder–dataset pair, **bold** denotes the highest accuracy and the lowest bias. Values are means over 5 seeds; standard deviations are provided in Appendix D.2.

| Encoder | Method | German | | | Credit | | | Bail | | |
|---|---|---|---|---|---|---|---|---|---|---|
| | | ACC↑ | SP↓ | EO↓ | ACC↑ | SP↓ | EO↓ | ACC↑ | SP↓ | EO↓ |
| GCN | Vanilla | 70.00 | 0.17 | 0.19 | 74.94 | 8.18 | 6.50 | **86.15** | 5.50 | 3.32 |
| | EST+FL | **70.16** | **0.08** | **0.15** | **77.86** | **0.03** | **0.03** | 85.36 | **0.85** | **0.36** |
| | EST-only | 69.92 | 0.13 | 0.17 | 74.96 | 4.32 | 3.20 | 84.92 | 0.96 | 0.84 |
| | FL-only | 70.00 | 0.17 | 0.19 | 77.58 | 0.40 | 0.37 | 82.14 | 1.72 | 0.79 |
| GIN | Vanilla | 71.04 | 3.36 | 1.26 | 75.73 | 1.18 | 0.66 | 84.81 | 7.27 | 6.55 |
| | EST+FL | 70.08 | **0.21** | **0.17** | **78.26** | **0.05** | 0.31 | 82.71 | **1.70** | **0.94** |
| | EST-only | 71.04 | 0.72 | 0.67 | 77.91 | 0.50 | **0.28** | 82.54 | 2.42 | 2.62 |
| | FL-only | **72.08** | 1.10 | 1.62 | 77.72 | 0.93 | 1.11 | **85.12** | 7.50 | 6.39 |
| SAGE | Vanilla | 70.00 | 0.85 | 0.71 | 74.30 | 8.68 | 6.44 | 88.21 | **0.67** | 1.92 |
| | EST+FL | 71.28 | **0.25** | 0.15 | **77.86** | **0.06** | **0.12** | 87.44 | 2.46 | 2.63 |
| | EST-only | **71.76** | 1.19 | 0.19 | 74.14 | 2.37 | 0.42 | 87.64 | 2.55 | 3.17 |
| | FL-only | 70.80 | 0.67 | **0.11** | 76.21 | 2.45 | 2.03 | **88.29** | 0.87 | **1.32** |

## 6.4 ABLATION STUDY

We study individual contributions of the FairEST components—the pre-processing algorithm (EST) and the fairness loss (FL)—by comparing four variants: (i) Vanilla (no EST, no FL), (ii) EST+FL, (iii) EST-only, and (iv) FL-only. We report accuracy and fairness ($\Delta$SP and $\Delta$EO).

Table 2 shows that EST+FL delivers the strongest and most consistent bias reduction across encoder-dataset pairs. In contrast, EST-only and FL-only exhibit clear failure cases—e.g., on GCN-Credit, EST-only yields 7% higher bias than EST+FL, and on GIN-Bail, FL-only is 11% worse. Although each component typically improves over Vanilla, neither is reliably effective in isolation. In terms of utility, EST-only often degrades accuracy, whereas FL-only tends to improve it. Their combination balances these effects, with EST+FL typically matching Vanilla on accuracy. Overall, EST and FL are complementary—both are needed for robust debiasing with comparable utility.

## 6.5 NOISY SENSITIVE ATTRIBUTES

Following prior research, we assume $s$ is available for all nodes during training (Yang et al., 2024). Nevertheless, we corrupt $s$ on val/test nodes with error rate $\epsilon$, to assess robustness to noisy sensitive attributes. Note that we fix $k = 0$ to isolate the effect of attribute noise with other features.

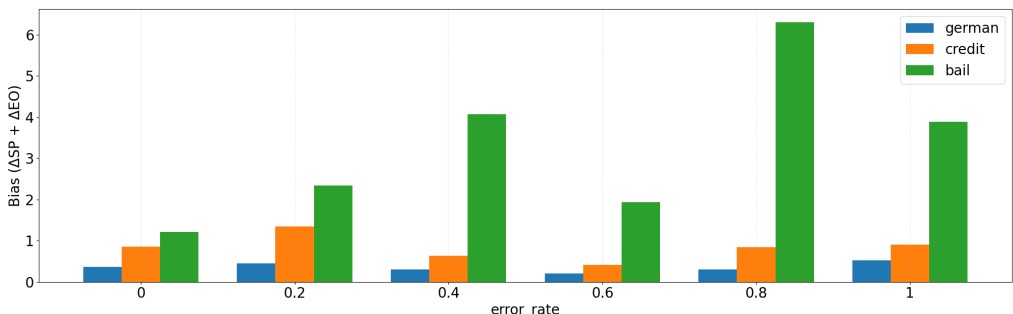

Figure 3: Composite bias ($\Delta$SP + $\Delta$EO) vs. sensitive-attribute error rate $\epsilon$ with a GCN backbone. Grouped bars correspond to German, Credit, and Bail. On German/Credit, bias remains $\leq 1.5\%$ with minimal variation across $\epsilon$, while Bail exhibits larger fluctuations yet still performs better than the second-best debiasing baseline on GCN-Bail (7.79% for NIFTY).

Figure 3 plots composite bias ($\Delta$SP + $\Delta$EO) with respect to sensitive-attribute error rate $\epsilon$ under a GCN backbone. On German and Credit, bias remains $\leq 1.5\%$ with minimal variation across $\epsilon$, implying that FairEST continues to suppress bias under noisy sensitive attributes. Bail exhibits larger fluctuations, yet the worst case ($\epsilon = 0.8$) outperforms the second-best debiasing baseline on GCN-Bail (7.79% for NIFTY). These results suggest that FairEST's primary goal—enforcing $h_s \approx 0.5$—can be achieved even when the sensitive attribute is erroneous on val/test nodes. Consequently, this supports the practicality of FairEST when $s$ must be predicted or is imperfectly observed.

### 6.6 LIMITATIONS AND FUTURE WORK

Despite the successful bias reduction of FairEST, several limitations remain. (i) We focus on binary labels and a single binary sensitive attribute. Extending to multi-class tasks and multiple sensitive attributes would broaden applicability. (ii) FairEST can shift the marginal $P(s)$, which may in turn influence bias. Incorporating base-rate priors to preserve $P(s)$ is a promising direction. (iii) FairEST operates on features $X$ while keeping the adjacency $A$ fixed. Fairness-aware edge rewiring or joint optimization over $(X, A)$ may yield additional gains.

## 7 CONCLUSION

We revisited fairness in GNNs through the lens of homophily. Using **CSBM-S**, a synthetic graph model, we observed that group disparity peaks when label homophily $h_y \approx 0.5$, whereas steering sensitive homophily toward $h_s \rightarrow 0.5$ suppresses bias. Building on these insights, we proposed **FairEST**, a fairness-aware GNN that enforces $h_s \approx 0.5$ by iteratively adjusting sensitive attributes and their most correlated feature dimensions. Across diverse backbones and benchmarks, FairEST substantially reduces $\Delta$SP and $\Delta$EO while matching the accuracy of vanilla GNNs, indicating that shaping sensitive homophily can improve group fairness without sacrificing utility. We expect CSBM-S and FairEST to facilitate future research and encourage homophily-aware evaluations of fairness in graph representation learning.

**Reproducibility statement.** Implementation details—model architectures, hyperparameters, and data splits—are provided in Appendix C. Prior debiasing methods (Section 2) are implemented from their official GitHub repositories. Dataset statistics appear in Appendix B.2, and evaluation metrics are defined in Section 3.3. All experiments are run across 5 random seeds, with means reported in the main text and standard deviations in Appendix D. Source code and scripts to reproduce CSBM-S and FairEST are included in the supplementary materials.

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

Table 3: Dataset statistics for German, Credit, and Bail.

| Dataset | $|\mathcal{V}|$ | $|\mathcal{E}|$ | $d$ | $\mathcal{H}_y$ | $\mathcal{H}_s$ |
|---------|------|------|------|------|------|
| German | 1,000 | 22,242 | 27 | 0.60 | 0.81 |
| Credit | 30,000 | 1,436,858 | 13 | 0.74 | 0.96 |
| Bail | 18,876 | 321,308 | 18 | 0.78 | 0.54 |

## A  ADDITIONAL OBSERVATIONS

Beyond the main observations discussed in Section 4.4, Figure 1(b) reveals two additional patterns: (i) $\Delta$SP is typically larger than $\Delta$EO: most $\Delta$EO values are below 3%, whereas $\Delta$SP values are often near 10%; and (ii) the label homophily $h_y$ that minimizes each metric differs: $\Delta$SP decreases as $h_y \to 0.5$ (even though it peaks at $h_y = 0.5$), while $\Delta$EO attains its minimum under extreme label homophily (e.g., $h_y \approx 0.1$ or $0.9$).

These behaviors explain how each metric responds to high accuracy—typically under extreme $h_y$. Assuming near-perfect predictions $\hat{y} \approx y$:

**Statistical parity.** From the definition of statistical parity (Eq. 3),

$$P(\hat{y}=1|s=0) \approx P(y=1|s=0) = \frac{P(y=1, s=0)}{P(s=0)} = \frac{(1-\rho)/4}{1/2} = \frac{1-\rho}{2},$$
$$P(\hat{y}=1|s=1) \approx P(y=1|s=1) = \frac{P(y=1, s=1)}{P(s=1)} = \frac{(1+\rho)/4}{1/2} = \frac{1+\rho}{2}. \tag{19}$$

Hence,

$$\Delta\text{SP} = |P(\hat{y}=1|s=0) - P(\hat{y}=1|s=1)| \approx \left| \frac{1-\rho}{2} - \frac{1+\rho}{2} \right| = |\rho|. \tag{20}$$

Thus, when predictions are accurate, $\Delta$SP is governed by the inherent correlation between $y$ and $\mathbf{s}$, explaining the relatively larger $\Delta$SP values observed in high-accuracy regimes.

**Equal opportunity.** From the definition of equal opportunity (Eq. 4),

$$\Delta\text{EO} = |P(\hat{y}=1|y=1, s=0) - P(\hat{y}=1|y=1, s=1)| \approx |1 - 1| = 0, \tag{21}$$

since $\hat{y} \approx y = 1$ under near-perfect predictions, regardless of $s$. Consequently, $\Delta$EO is minimized in high-accuracy regimes (e.g., under extreme $h_y$), consistent with Figure 1(b).

## B  DATASET DETAILS

### B.1  CSBM-S EXPERIMENT

We generate CSBM-S graphs with $n = 10,000$ nodes and a label-sensitive correlation $\rho = 0.1$. Node features are sampled from Gaussians with symmetric means $\mu_{y1} = -\mu_{y0}$, $\mu_{s1} = -\mu_{s0}$ and unit variances $\sigma_{y0} = \sigma_{y1} = \sigma_{s0} = \sigma_{s1} = 1$. To strictly enforce a sensitive-balanced neighborhood for every node, we use directed graphs with fixed in-degree $d = 100$.

### B.2  MAIN EXPERIMENT

Table 3 summarizes the dataset statistics used in the main experiments—German, Credit, and Bail. For each dataset, we report the number of nodes ($|\mathcal{V}|$), edges ($|\mathcal{E}|$), and input features ($d$), along with label homophily ($\mathcal{H}_y$), and sensitive homophily ($\mathcal{H}_s$). Note that homophily is measured as edge homophily—the fraction of edges connecting nodes with the same label (for $\mathcal{H}_y$) or the same sensitive attribute (for $\mathcal{H}_s$)

## C  IMPLEMENTATION DETAILS

### C.1  ENVIRONMENT

Experiments were conducted on a server with AMD Ryzen Threadripper PRO 5975WX (1.8GHz), 1TB DDR4 RAM, and an NVIDIA GeForce RTX 4090 (CUDA 12.8), running Ubuntu 24.04.1 LTS. Software versions include Python 3.8.19 and PyTorch 2.4.1.

### C.2  CSBM-S EXPERIMENT

We implement CSBM-S by modifying the CSBM-X reference code of Lee et al. (2024). The encoder is a 1-layer simplified GCN with a linear classification head. Models are trained for up to 500 epochs with early stopping (patience $= 100$ based on validation loss). We use a $50/25/25$ train/val/test split, learning rate $0.01$, and weight decay $5 \times 10^{-4}$.

### C.3  MAIN EXPERIMENT

We detail architectures and training hyperparameters used for FairEST. Other debiasing baselines use the authors' released architectures and hyperparameters unless noted.

**Encoders.** The GCN encoder applies one linear layer followed by one GCN layer. The GIN encoder consists of one GIN convolution, then a linear layer with batch normalization. The SAGE encoder uses one SAGE convolution layer followed by ReLU, batch normalization, and dropout ($p = 0.5$). Note that all encoders feed a 1-layer MLP classifier.

**Training.** We train for $\{200, 300, 400\}$ outer epochs, and within each outer epoch, the classifier receives $\{5, 10\}$ updates. Learning rates for encoder and classifier are selected from $\{0.001, 0.01\}$. Hidden dimension (encoder output / classifier input) is 16. The checkpoint used for inference is the one with the highest composite validation score (ACC + AUC + F1 $- \Delta$SP $- \Delta$EO).

**Data splits.** Train/val/test node counts are German: $100/250/250$; Credit: $6000/7500/7500$; Bail: $100/4719/4719$.

**Hyperparameters.** We tune methods by maximizing the combined objective ACC $- \Delta$SP $- \Delta$EO.

- **FairGNN.** Hidden dim $= 32$; learning rate $\in \{0.0001, 0.001, 0.01\}$; weight decay $= 10^{-5}$; dropout $\in \{0.0, 0.5, 0.8\}$. Regularization coefficients $\alpha = 4, \beta = 0.01$. Sensitive number and label number: 200 and 500.

- **NIFTY.** Hidden dim $= 16$; projection hidden dim $= 16$; weight decay $\in \{10^{-5}, 10^{-4}\}$; learning rate $\in \{0.0001, 0.001, 0.01\}$; dropout $\in \{0.0, 0.5, 0.8\}$. Drop edge rate $= 0.001$; drop feature rate $= 0.1$. Regularization coefficient $\in \{0.4, 0.5, 0.6, 0.7, 0.8\}$.

- **EDITS.** Learning rate $= 0.003$; weight decay $= 10^{-7}$. Threshold proportions: German $= 0.25$, Credit $= 0.02$, Bail $= 0.012$.

- **FairVGNN, FairSIN.** Authors' implementations with recommended hyperparameters.

- **FairEST.** Number of correlated feature dimensions $k \in \{0, 1, 2, 3, 4\}$. Fairness loss term coefficient $\lambda \in \{0.01, 0.1, 1, 10, 100\}$.

## D  STANDARD DEVIATION

### D.1  MAIN RESULTS

Table 4 reports standard deviations (over 5 seeds) for the main results in Section 6.2.

### D.2  ABLATION STUDY

Table 5 reports standard deviations (over 5 seeds) for the ablation results in Section 6.4.

Table 4: Standard deviations (over 5 seeds) for Section 6.2 results.

| Encoder | Method | German | | | Credit | | | Bail | | |
|---|---|---|---|---|---|---|---|---|---|---|
| | | ACC | SP | EO | ACC | SP | EO | ACC | SP | EO |
| GCN | Vanilla | 0.00 | 0.34 | 0.38 | 1.52 | 4.90 | 4.17 | 0.50 | 0.60 | 0.28 |
| | FairGNN | 0.16 | 0.41 | 0.73 | 3.34 | 0.24 | 0.50 | 1.14 | 0.60 | 1.31 |
| | NIFTY | 0.90 | 1.19 | 0.83 | 3.65 | 3.84 | 2.72 | 1.02 | 0.80 | 0.81 |
| | EDITS | 0.59 | 2.02 | 1.17 | 2.21 | 3.46 | 3.66 | 12.48 | 4.90 | 3.24 |
| | FairVGNN | 0.41 | 1.71 | 1.71 | 0.42 | 2.76 | 1.34 | 0.48 | 0.33 | 0.67 |
| | FairSIN | 0.41 | 0.58 | 0.62 | 0.02 | 0.57 | 0.33 | 0.34 | 0.52 | 0.87 |
| | FairEST | 0.32 | 0.16 | 0.29 | 0.04 | 0.05 | 0.06 | 0.91 | 0.71 | 0.16 |
| GIN | Vanilla | 0.82 | 2.26 | 1.24 | 2.87 | 0.74 | 0.53 | 0.90 | 0.43 | 0.46 |
| | FairGNN | 1.06 | 1.72 | 0.37 | 1.67 | 0.53 | 0.70 | 0.30 | 0.44 | 0.43 |
| | NIFTY | 0.00 | 0.17 | 0.34 | 1.95 | 2.87 | 2.12 | 2.60 | 0.59 | 0.72 |
| | EDITS | 0.86 | 2.22 | 0.39 | 0.18 | 1.43 | 1.29 | 15.94 | 3.32 | 2.93 |
| | FairVGNN | 1.86 | 4.61 | 1.71 | 0.46 | 0.43 | 0.34 | 0.21 | 0.34 | 0.68 |
| | FairSIN | 0.89 | 1.89 | 1.36 | 0.01 | 0.11 | 0.07 | 0.63 | 1.14 | 0.76 |
| | FairEST | 0.16 | 0.27 | 0.34 | 0.76 | 0.10 | 0.62 | 2.06 | 1.01 | 1.21 |
| SAGE | Vanilla | 0.25 | 1.24 | 1.43 | 3.35 | 4.61 | 3.99 | 0.52 | 0.31 | 0.77 |
| | FairGNN | 0.78 | 0.48 | 1.48 | 1.71 | 1.76 | 1.02 | 0.22 | 0.50 | 0.74 |
| | NIFTY | 0.16 | 0.40 | 1.39 | 1.15 | 2.44 | 2.47 | 3.61 | 2.01 | 1.32 |
| | EDITS | 0.74 | 4.72 | 1.87 | 0.02 | 0.03 | 0.04 | 14.10 | 3.05 | 4.42 |
| | FairVGNN | 0.76 | 3.22 | 1.28 | 0.87 | 3.80 | 2.61 | 0.44 | 0.88 | 1.27 |
| | FairSIN | 1.09 | 5.18 | 2.73 | 0.74 | 2.19 | 0.97 | 0.36 | 0.71 | 0.70 |
| | FairEST | 1.42 | 0.42 | 0.24 | 0.13 | 0.09 | 0.17 | 1.02 | 2.44 | 1.84 |

Table 5: Standard deviations (over 5 seeds) for Section 6.4 ablations.

| Encoder | Method | German | | | Credit | | | Bail | | |
|---|---|---|---|---|---|---|---|---|---|---|
| | | ACC | SP | EO | ACC | SP | EO | ACC | SP | EO |
| GCN | Vanilla | 0.00 | 0.34 | 0.38 | 1.52 | 4.90 | 4.17 | 0.50 | 0.60 | 0.28 |
| | EST+FL | 0.32 | 0.16 | 0.29 | 0.04 | 0.05 | 0.06 | 0.91 | 0.71 | 0.16 |
| | EST-only | 0.16 | 0.26 | 0.34 | 3.89 | 4.07 | 3.30 | 0.66 | 0.68 | 0.56 |
| | FL-only | 0.00 | 0.34 | 0.38 | 0.50 | 0.50 | 0.45 | 0.40 | 0.10 | 0.11 |
| GIN | Vanilla | 0.82 | 2.26 | 1.24 | 2.87 | 0.74 | 0.53 | 0.90 | 0.43 | 0.46 |
| | EST+FL | 0.16 | 0.27 | 0.34 | 0.76 | 0.10 | 0.62 | 2.06 | 1.01 | 1.21 |
| | EST-only | 0.86 | 0.94 | 0.63 | 0.05 | 1.01 | 0.56 | 1.42 | 1.15 | 1.41 |
| | FL-only | 1.46 | 0.85 | 1.28 | 1.35 | 0.78 | 0.61 | 1.60 | 0.46 | 0.41 |
| SAGE | Vanilla | 0.25 | 1.24 | 1.43 | 3.35 | 4.61 | 3.99 | 0.52 | 0.31 | 0.77 |
| | EST+FL | 1.42 | 0.42 | 0.24 | 0.13 | 0.09 | 0.17 | 1.02 | 2.44 | 1.84 |
| | EST-only | 1.44 | 1.19 | 0.38 | 3.52 | 1.20 | 0.25 | 0.55 | 1.83 | 2.25 |
| | FL-only | 1.10 | 1.24 | 0.21 | 1.86 | 2.38 | 2.07 | 0.97 | 0.68 | 1.12 |

# E   USE OF LARGE LANGUAGE MODELS (LLMS)

We used LLMs solely for aiding and polishing writing. All technical content, experiments, analyses, and conclusions were produced by the authors. All suggested edits were reviewed and verified for correctness and consistency prior to inclusion.

