# OpenReview forum: "Revisiting the Role of Homophily in Fair Graph Representation Learning"
_ICLR.cc/2026/Conference — ICLR 2026 Conference Withdrawn Submission_

### Official Review · Reviewer_Wbsp · 2025-10-18

**Soundness:** 3
**Presentation:** 2
**Contribution:** 2
**Rating:** 4
**Confidence:** 4

**Summary:**

The authors introduce CSBM-S, a controllable synthetic benchmark that decouples label homophily (h_y) and sensitive homophily (h_s), enabling precise evaluation of fairness mechanisms.
CSBM-S identifies two empirical trends: Group disparity peaks when label homophily and Bias tend to decrease as sensitive homophily
Then, based on these insights, they propose FairEST, a method that enforces ~0.5 by flipping sensitive attributes and correlated features during training to mitigate bias. Experimental results show consistent improvements in fairness metrics across baselines.

**Strengths:**

I like the idea of fairness in GNNs through a homophily view, offering a new conceptual angle on how topology affects bias propagation. Also, FairEST is conceptually straightforward, model-agnostic, and integrates easily into existing GNN pipelines.
The observations linking fairness with specific homophily ranges could inform future fairness-aware graph design.

**Weaknesses:**

- To me, GNNs inherently rely on the homophily principle, learning from neighboring nodes under propagation. Therefore, attributing fairness issues primarily to homophily may oversimplify the problem. The root causes of unfairness might instead come from global structural factors, such as community topology or node identity, rather than local structural properties like node degree or neighborhood similarity.

- The rationale for flipping sensitive attributes may appear heuristic. That is, its connection to causality or representation disentanglement could be better articulated.

**Questions:**

- Could the authors clarify whether the feature flipping operation might leak or distort semantic information critical for downstream tasks?

- How does FairEST perform on heterophilous graphs where h_y and h_s  are both low? Does the method still yield fairness gains?

- How sensitive is FairEST to incorrect or noisy sensitive attributes?

---

> ### Author Response · Authors · 2025-11-18
>
> Thank you for your thoughtful and constructive feedback. We address the main points below.
>
> ### 1. Could the authors clarify whether the feature flipping operation might leak or distort semantic information critical for downstream tasks?
>
> We agree that potential leakage or distortion of semantic information is an important concern.
> In our experiments, this effect is indirectly reflected in the accuracy changes induced by FairEST.
> Compared to the vanilla model, accuracy is typically improved on German and Credit, while it experiences a drop on Bail.
> Since there is an inherent trade-off between accuracy and fairness, it is difficult to optimize both objectives simultaneously.
> However, FairEST makes this trade-off explicitly controllable via the hyperparameter $k$, which determines how many feature dimensions are flipped jointly.
> By tuning $k$, we can adjust the strength of the feature flipping and thereby control the extent to which semantic information is perturbed, depending on the requirements of the downstream task (Section 6.3).
>
> ### 2. How does FairEST perform on heterophilous graphs where h_y and h_s are both low? Does the method still yield fairness gains?
>
> While it is difficult to evaluate FairEST on real-world heterophilous benchmarks (since most existing datasets are homophilous), we can infer its behavior from the CSBM-S experiments.
> Figure 1 shows that not only high $h_s$ but also very low $h_s$ lead to biased predictions.
> In other words, deviations of $h_s$ in either direction (i.e., $h_s \gg 0.5$ or $h_s \ll 0.5$) increase bias, and pushing $h_s$ toward 0.5 reduces bias.
> Since FairEST is designed to move neighborhood-level $h_s$ toward 0.5, it is expected to provide fairness gains in both homophilous and heterophilous regimes of $s$.
> $h_y$ influences how much these gains translate into fairness improvements: when $h_y$ is extremely low (e.g., $h_y \approx 0.1$), the effect of FairEST decreases, because the classifier places more weight on $y$ than on $s$.
> We believe that constructing "sensitive but heterophilous" benchmark datasets is an important direction for future work in fair graph representation learning.
>
> ### 3. How sensitive is FairEST to incorrect or noisy sensitive attributes?
>
> We agree that in high-stakes settings $s$ is often hidden or unavailable and should be predicted from proxies, which introduces noise into $s$.
> To address this, Section 6.5 analyzes the robustness of TIER under noisy sensitive labels.
> For German and Credit, TIER exhibits only minimal fluctuation as the error rate increases, indicating strong robustness to noise in $s$.
> For Bail, TIER shows somewhat larger fluctuations under noisy $s$, but even in the worst case it remains less biased than prior adversarial/invariant approaches [1, 2].
> These results suggest that TIER’s main objective—enforcing neutral sensitive homophily—can still be achieved in realistic settings where $s$ is not directly observed and must be predicted from proxies.
>
> ### References
> [1] Dai & Wang "Say no to the discrimination: Learning fair graph neural networks with limited sensitive attribute information"
>
> [2] Wang et al. "Improving fairness in graph neural networks via mitigating sensitive attribute leakage"

---

### Official Review · Reviewer_URUk · 2025-10-21

**Soundness:** 2
**Presentation:** 2
**Contribution:** 1
**Rating:** 2
**Confidence:** 5

**Summary:**

This work aims to study fairness in GNNs from the perspective of homophily. Specifically, the authors focus on notions of label and sensitive attribute homophily, assessing which neighborhood patterns cause fairness degradation. Through their CSBM-S model, a synthetic graph model that controls the label and sensitive attribute homophily, the authors demonstrate that group fairness degrades as label homophily tends towards 0.5, while group fairness improves as sensitive attribute homophily tends towards 0.5. To use these findings, the authors present FairEST, a method which aims to optimize the sensitive attribute homophily at training-time to 0.5. Generally, FairEST is able to achieve decent fairness metrics, but does incur a performance cost.

**Strengths:**

1. I found the method sections of the paper relatively easy to read given that each section naturally follows from the previous and logically builds on one another.
2. The empirical results, at least for the fairness metrics, tend to be decent.
3. The methods simplicity as a training-time augmentation makes it amenable to different backbones and settings.

**Weaknesses:**

1. My main concern about this work is that it largely uses methods and insights already well established in the literature. Moreover, the authors do not offer enough arguments as to how their work explicitly differs from these methods, sometimes missing citations all together. A few examples are:

    - Homophily and fairness are already well connected in the literature. As far as I can tell, the authors do not explicitly address how their work builds on, or "revisits", these previous findings [1, 2, 3].
    - Beyond just connecting homophily and fairness, the proposed CSBM-S model and analysis in section 4 are highly similar to that in [2], both from the model design of manipulating label and sensitive attribute homophily, as well as the resulting takeaways.
    - The idea of flipping the sensitive attribute, aiming to "debias" the message passing process, does not seem sufficiently different from previous methods which manipulate the graph structure to encourage different treatments across sensitive attributes [3, 4].

2. For the majority of the experimental results, while the fairness metrics are decent, the accuracy drops are sometimes quite large. Given my points above, I think significantly more effort needs to go in to remedy this issue and establish more novelty in the method.

3. While trying to assess whether the author justified the performance drops, I realized there are instances in section 6.2 which do not seem to correspond to Table 1. For instance, on line 364, the reported accuracy numbers changes are significantly higher than the drops seen in the table (e.g., authors report -2.1% on GIN-bail, yet it would appear the drop is closer to 5.5% in table 1).

In all, I think this work needs quite a bit more effort to both sufficiently ground itself in the literature and also improve presentation in the experimental section.

[1] Wang et al. “Improving Fairness in Graph Neural Networks via Mitigating Sensitive Attribute Leakage”
[2] Loveland et al. “On Graph Neural Network Fairness in the Presence of Heterophilous Neighborhoods”
[3] Li et al. “On Dyadic Fairness: Exploring and Mitigating Bias in Graph Connections”
[4] Rahman et al. “Fairwalk: Towards Fair Graph Embedding”

**Questions:**

Please see my weaknesses above.

---

> ### Author Response · Authors · 2025-11-18
>
> Thank you for your thoughtful and constructive feedback. We address the main points below.
>
> ### 1. My main concern about this work is that it largely uses methods and insights already well established in the literature. Moreover, the authors do not offer enough arguments as to how their work explicitly differs from these methods, sometimes missing citations all together.
>
> First, while prior works [1, 2, 3] do suggest that high homophily can contribute to bias, they do not systematically characterize the regimes of $h_y$ and $h_s$ under which bias is amplified.
> This is precisely the main analytical contribution of our paper.
> For instance, [1] argues that decreasing homophily can smooth out sensitive information, but does not study the opposite extreme where very low homophily can again increase bias—an effect we explicitly identify and analyze.
> Second, our synthetic model (CSBM-S) differs from that of [2]: (i) we allow explicit control over the correlation between $y$ and $s$, an important driver of bias (even if we do not fully exploit this axis in the current paper), and (ii) [2] relies on probabilistic edge sampling, which may fail to reveal certain homophily-fairness relationships (e.g., bias peaking exactly at $h_y=0.5$).
> Moreover, [2] primarily uses its synthetic model to compare homophilous vs. heterophilous GNN architectures, rather than to study the homophily-fairness relationship itself.
> Third, adjacency-adjustment methods [3, 4] operate directly on the graph structure and therefore risk discarding task-related information, since edges encode both $s$ and $y$.
> In contrast, TIER intervenes only on sensitive-related features, with intervention strength controlled by the hyperparameter $k$.
> This design allows TIER to mitigate bias while better preserving task-related structural information.
>
> ### 2. For the majority of the experimental results, while the fairness metrics are decent, the accuracy drops are sometimes quite large. Given my points above, I think significantly more effort needs to go in to remedy this issue and establish more novelty in the method.
>
> We agree that accuracy drops can be non-negligible in some settings, especially on Bail.
> However, compared to the vanilla model, accuracy is typically improved on German and Credit, while it decreases on Bail.
> Since there is an inherent trade-off between accuracy and fairness, optimizing both objectives simultaneously is fundamentally challenging.
> A key advantage of FairEST is that this trade-off is explicitly controllable via the hyperparameter $k$, which governs how many feature dimensions are flipped jointly.
> By tuning $k$, we can adjust the strength of the intervention and thereby control how much semantic information is perturbed, depending on the requirements of the downstream task (Section 6.3).
>
> ### 3. While trying to assess whether the author justified the performance drops, I realized there are instances in section 6.2 which do not seem to correspond to Table 1. For instance, on line 364, the reported accuracy numbers changes are significantly higher than the drops seen in the table (e.g., authors report -2.1% on GIN-bail, yet it would appear the drop is closer to 5.5% in table 1).
>
> We thank the reviewer for carefully checking these numbers and for pointing out the inconsistency.
> The performance drops reported on line 364 were computed relative to the vanilla models (as stated on line 363), whereas the bias comparisons on line 358 are made against FairSIN.
> We acknowledge that this was not sufficiently emphasized and can indeed be confusing when read alongside Table 1, where metrics are presented in a different comparative format.
> We did not intend to understate the accuracy drops; we chose the vanilla models as the reference point because we view bias reduction as the primary objective of fairness-aware GNNs, with utility as a secondary (yet still important) consideration.
> In the revised version, we will (i) clearly state the reference baselines used for both accuracy and fairness in Section 6.2, and (ii) ensure that all reported performance drops are fully consistent with Table 1.
> More broadly, we recognize that accuracy drops and distributional shifts are inherent in FairEST, and we plan to investigate this trade-off more systematically in future work.
>
> ### References
> [1] Wang et al. "Improving Fairness in Graph Neural Networks via Mitigating Sensitive Attribute Leakage"
>
> [2] Loveland et al. "On Graph Neural Network Fairness in the Presence of Heterophilous Neighborhoods"
>
> [3] Li et al. "On Dyadic Fairness: Exploring and Mitigating Bias in Graph Connections"
>
> [4] Rahman et al. "Fairwalk: Towards Fair Graph Embedding"

---

### Official Review · Reviewer_3Sxf · 2025-10-23

**Soundness:** 3
**Presentation:** 3
**Contribution:** 3
**Rating:** 6
**Confidence:** 4

**Summary:**

The paper studies group fairness in GNNs through the lens of label homophily $h_y$ and sensitive homophily $h_s$. It introduces CSBM-S, a synthetic generator that independently controls $h_y$ and $h_s$ to analyze bias under message passing, observing that disparity peaks near $h_y \approx 0.5$ and diminishes as $h_s \to 0.5$. Building on this, the authors propose FairEST, which iteratively edits the sensitive attribute $s$ and its most correlated features to steer neighborhoods toward $h_s \approx 0.5$, with an auxiliary group-fairness loss. Experiments across multiple datasets and GNN backbones show reduced group-fairness gaps with comparable accuracy.

**Strengths:**

+ The motivation is clear. The paper formalizes node-level $h_y$, $h_s$, and standard group-fairness metrics ($\Delta\mathrm{SP}$, $\Delta\mathrm{EO}$), then analyzes how message passing amplifies or attenuates disparities. It further employs CSBM-S to vary $h_y$ and $h_s$ independently. Grid sweeps and a mean-field analysis yield interpretable patterns that motivate the method.
+  FairEST is backbone-agnostic and easy to implement. Edits to $s$ and correlated features, combined with a fairness loss, reduce bias without architectural changes.
+ The experimental study is extensive, covering multiple datasets/backbones with ablations and hyperparameter analyses that reveal both gains and failure modes.

**Weaknesses:**

- The method assumes the sensitive attribute $s$ is observed, which may be unrealistic in high-stakes settings. Please evaluate fairness when $s$ is hidden or unavailable, e.g., by comparing to adversarial/invariant approaches and to a setting where $s$ is predicted from proxies.
-  The algorithm greedily balances neighborhood homophily $h_s$ toward $0.5$ using node-wise majority and a fixed iteration cap. It is unclear whether these local flips provably reduce global disparity or instead induce distributional shifts in $P(s)$.
- The paper currently targets binary labels and a single binary sensitive attribute. How about the generalization to multi-class or multi-attribute settings?
-  Only group fairness is evaluated. Individual fairness is discussed conceptually but not assessed. Many applications may require both.

**Questions:**

Please refer to the above weaknesses.

---

> ### Author Response · Authors · 2025-11-18
>
> Thank you for your thoughtful and constructive feedback. We address the main points below.
>
> ### 1. The method assumes the sensitive attribute $s$ is observed, which may be unrealistic in high-stakes settings. Please evaluate fairness when $s$ is hidden or unavailable, e.g., by comparing to adversarial/invariant approaches and to a setting where $s$ is predicted from proxies.
>
> We agree that in high-stakes settings $s$ is often hidden or unavailable and should be predicted from proxies, which introduces noise into $s$.
> To address this, Section 6.5 analyzes the robustness of TIER under noisy sensitive labels.
> For German and Credit, TIER exhibits only minimal fluctuation as the error rate increases, indicating strong robustness to noise in $s$.
> For Bail, TIER shows somewhat larger fluctuations under noisy $s$, but even in the worst case it remains less biased than prior adversarial/invariant approaches [1, 2].
> These results suggest that TIER’s main objective—enforcing neutral sensitive homophily—can still be achieved in realistic settings where $s$ is not directly observed and must be predicted from proxies.
>
> ### 2. The algorithm greedily balances neighborhood homophily $h_s$ toward $0.5$ using node-wise majority and a fixed iteration cap. It is unclear whether these local flips provably reduce global disparity or instead induce distributional shifts in $P(s)$.
>
> | Metric | FairEST  | German | Credit | Bail |
> |--------|----------|--------|--------|------|
> | P(s=0) | before   | 0.69   | 0.91   | 0.49 |
> | P(s=0) | after    | 0.51   | 0.50   | 0.50 |
> | $h_s$  | before   | 0.81   | 0.96   | 0.54 |
> | $h_s$  | after    | 0.50   | 0.50   | 0.50 |
>
> The table above reports the global sensitive distribution $P(s)$ and sensitive homophily $h_s$ before and after applying FairEST.
> We observe that, although FairEST operates through local flips, these local operations also reduce global disparity: $h_s$ moves toward 0.5 for all datasets.
> This indicates that the local majority-based updates do, in practice, yield the intended global effect.
> At the same time, FairEST induces distributional shifts in $P(s)$ toward 0.5, as discussed in Section 6.6.
> A more balanced $P(s)$ (e.g., $P(s=0) \approx 0.5$) can further improve group fairness, but it may also introduce a mismatch between the training and test distributions, potentially harming generalization.
> Thus, FairEST’s local flips successfully enforce more similar neighborhoods while modestly shifting $P(s)$; formally characterizing and controlling this trade-off is an important direction for future work.
>
> ### 3. The paper currently targets binary labels and a single binary sensitive attribute. How about the generalization to multi-class or multi-attribute settings?
>
> We chose the binary setting primarily to align with widely used benchmark datasets and fairness metrics, and to keep the exposition focused.
> However, the proposed framework can naturally be extended to multi-class and multi-attribute settings.
> First, fairness and homophily quantities can be defined separately for each class and each sensitive group, so the same measurement principles apply beyond the binary case.
> Second, the main mechanism by which FairEST reduces bias is by making the aggregated sensitive information in each node’s neighborhood similar, so that the final classifier cannot exploit sensitive information for prediction.
> This mechanism does not rely on binary sensitive attributes: FairEST can be extended to multiple sensitive attributes by encouraging all neighborhoods to share similar (possibly high-dimensional) sensitive distributions.
> We will clarify this generalization to multi-class and multi-attribute settings in the revised version.
>
> ### 4. Only group fairness is evaluated. Individual fairness is discussed conceptually but not assessed. Many applications may require both.
>
> In this work, we deliberately focused on group fairness for two main reasons: (i) group-based metrics such as statistical parity and equal opportunity are the predominant evaluation standards in the fair GNN literature, and (ii) widely used benchmark datasets and existing methods are primarily designed around group fairness rather than individual fairness [1, 2, 3].
> Nevertheless, we agree that individual fairness in graph settings is an important and underexplored direction.
> We view extending both our synthetic graph model (CSBM-S) and our fairness-aware GNN (FairEST) to explicitly address individual fairness as a promising avenue for future work, and we will clarify this limitation and outlook in the revised version.
>
> ### References
> [1] Dai & Wang. "Say no to the discrimination: Learning fair graph neural networks with limited sensitive attribute information."
>
> [2] Wang et al. "Improving fairness in graph neural networks via mitigating sensitive attribute leakage."
>
> [3] Yang et al. "Fairsin: Achieving fairness in graph neural networks through sensitive information neutralization."

---

### Official Review · Reviewer_HiJw · 2025-10-31

**Soundness:** 2
**Presentation:** 2
**Contribution:** 2
**Rating:** 2
**Confidence:** 3

**Summary:**

The paper studies the relationship between homophily and fairness in GNNs. They claim that the degree of label homophily $h_y$ and sensitive homophily $h_s$ significantly impacts bias amplification during message passing. To analyze this, they propose CSBM-S, a synthetic graph model that decouples label and sensitive homophily, allowing controlled experiments. They further introduce FairEST, an algorithm that enforces $h_s \approx 0.5$ to improve fairness by iteratively flipping sensitive attributes and correlated features. Experiments on several benchmarks and GNN baselines show modest improvements in fairness metrics with comparable accuracy.

**Strengths:**

1. The paper identifies an intersection between fairness and graph homophily. They show how label homophily and sensitive homophily could shape fairness under the message passing of GNNs.

2. The paper introduces CSBM-S as a controlled simulator for fairness studies, which allows disentangling the effects of different homophily levels in a reproducible manner, potentially benefiting future research.

3. Experiments include multiple models and datasets, and the authors conduct ablations, sensitivity analyses, and noise robustness tests.

**Weaknesses:**

1. The idea that message passing propagates sensitive signals via edges with attribute correlation is well-established (Wang et al., 2022; Dong et al., 2022; Dai & Wang, 2021). The notion of balancing sensitive attribute distributions (making $h_s \approx 0.5$) is just a graph-level rephrasing of feature decorrelation or resampling. FairEST’s “flip and reflect” procedure is essentially a stochastic data augmentation trick, not a theoretically or algorithmically novel approach. As such, the manuscript somewhat overstates the conceptual novelty of the approach by framing it as a “homophily-centric fairness framework,” when the underlying idea remains relatively straightforward.

2. In Section 4.4, the analysis appears to restate known results from mean-field diffusion analysis. The authors find that bias is largest when label information is weak ($h_y \approx 0.5$) and when sensitive channels dominate (extreme $h_s$). While this observation is intuitively consistent, i.e., bias tends to increase when sensitive features drive predictive signals. It does not seem to offer new theoretical insight into why or how GNN architectures amplify fairness issues.

3. Despite proposing CSBM-S, the paper does not use it to uncover deeper causal or structural insights about fairness dynamics in graphs. The synthetic experiments are mainly limited to grid-sweep heatmaps and a few straightforward observations, without quantitative analyses of robustness, sensitivity to graph topology, or comparisons with alternative fairness mechanisms. As a result, the proposed “homophily-centric toolkit” currently functions more as a synthetic data generator than as a framework for deeper theoretical understanding.

**Questions:**

1. How does FairEST compare with trivial strategies such as random node feature shuffling, attribute dropout, or standard reweighting schemes?

2. Are the fairness improvements statistically significant across runs? Please report confidence intervals or p-values.

3. The method assumes full access to sensitive attributes during training and precise correlation estimation, which is rarely met in practical settings. How would FairEST operate under partial or uncertain sensitive attribute availability?

4. Have the authors tested on larger or more realistic graphs (e.g., OGB datasets)? The current experimental setup lacks scalability evidence.

5. Does enforcing $h_s \approx 0.5$ actually remove causal influence of the sensitive attribute, or does it merely mask correlations?

---

> ### Author Response · Authors · 2025-11-18
>
> Thank you for your thoughtful and constructive feedback. We address the main points below.
>
> ### 1. How does FairEST compare with trivial strategies such as random node feature shuffling, attribute dropout, or standard reweighting schemes?
>
> The key difference between FairEST and the mentioned "trivial" approaches lies in robustness across encoder-dataset pairs.
> Random node shuffling, attribute dropout, and standard reweighting can sometimes reduce bias, but they often degrade performance or fail to consistently improve fairness.
> In contrast, FairEST combines a principled preprocessing algorithm with a fairness loss (Section 6.4), and we observe that this combination reduces bias much more reliably across models and datasets.
>
> ### 2. Are the fairness improvements statistically significant across runs? Please report confidence intervals or p-values.
>
> Standard deviations over 5 runs for the Section 6.2 results and Section 6.4 ablations are currently reported in Appendix D due to space limitations.
> In the revised version, we will surface this information in Tables 1 and 2, and additionally report confidence intervals to more clearly quantify the significance of the fairness gains.
>
> ### 3. The method assumes full access to sensitive attributes during training and precise correlation estimation, which is rarely met in practical settings. How would FairEST operate under partial or uncertain sensitive attribute availability?
>
> We agree that in high-stakes settings $s$ is often hidden or unavailable and should be predicted from proxies, which introduces noise into $s$.
> To address this, Section 6.5 analyzes the robustness of TIER under noisy sensitive labels.
> For German and Credit, TIER exhibits only minimal fluctuation as the error rate increases, indicating strong robustness to noise in $s$.
> For Bail, TIER shows somewhat larger fluctuations under noisy $s$, but even in the worst case it remains less biased than prior adversarial/invariant approaches [1, 2].
> These results suggest that TIER’s main objective—enforcing neutral sensitive homophily—can still be achieved in realistic settings where $s$ is not directly observed and must be predicted from proxies.
>
> ### 4. Have the authors tested on larger or more realistic graphs (e.g., OGB datasets)? The current experimental setup lacks scalability evidence.
>
> Unfortunately, existing benchmarks for fairness-aware GNNs are limited to small graphs, which makes it difficult to directly demonstrate scalability on larger benchmarks such as OGB within the standard fair GNN evaluation setup.
> Instead, we provide a complexity-based argument and comparison to recent approaches.
> FairEST requires computing $h_s$ for each node, which induces an additional cost of $O(|V|+|E|)$.
> This is no larger than the standard message-passing complexity of GNNs, $O(|V|+|E|)$, and is incurred only once as a preprocessing step, making the overhead negligible relative to the overall training cost.
> FairVGNN [2] exhibits similar complexity due to edge-weight clamping, but also requires a separate training process for the discriminator, leading to longer wall-clock time in practice.
> FairSIN [3] similarly aggregates information from heterophilous neighborhoods, making its running time scale with both $|V|$ and $|E|$.
> In the revised version, we will include a more explicit analysis of time complexity and wall-clock runtimes to assess the scalability of FairEST.
>
> ### 5. Does enforcing $h_s \approx 0.5$ actually remove causal influence of the sensitive attribute, or does it merely mask correlations?
>
> We appreciate the reviewer's causal perspective.
> In this work, our goal is to reduce the observational influence of $s$ rather than to provide formal causal guarantees.
> By enforcing $h_s \approx 0.5$, we neutralize each node's exposure to sensitive information in its neighborhood, so that the learned embeddings after message passing become less sensitive to $s$.
> Empirically, this is reflected in the reduced SP and EO gaps in Table 1, which are standard proxies for the predictive influence of $s$ in the fair GNN literature [1, 2, 3].
> Hence, we do not claim that FairEST eliminates all possible causal pathways from $s$; a more explicit causal analysis is an interesting direction for future work.
>
> ### References
> [1] Dai & Wang. "Say no to the discrimination: Learning fair graph neural networks with limited sensitive attribute information."
>
> [2] Wang et al. "Improving fairness in graph neural networks via mitigating sensitive attribute leakage."
>
> [3] Yang et al. "Fairsin: Achieving fairness in graph neural networks through sensitive information neutralization."

---

### Note · Authors · 2025-11-18

I have read and agree with the venue's withdrawal policy on behalf of myself and my co-authors.